# DEEP LINEAR PROBE GENERATORS FOR WEIGHT SPACE LEARNING

**Jonathan Kahana, Eliahu Horwitz, Imri Shuval, Yedid Hoshen**
School of Computer Science and Engineering
The Hebrew University of Jerusalem, Israel
`jonathan.kahana@mail.huji.ac.il`

Project page: **`https://vision.huji.ac.il/probegen/`**

## ABSTRACT

Weight space learning aims to extract information about a neural network, such as its training dataset or generalization error. Recent approaches learn directly from model weights, but this presents many challenges as weights are high-dimensional and include permutation symmetries between neurons. An alternative approach, *Probing*, represents a model by passing a set of learned inputs (probes) through the model, and training a predictor on top of the corresponding outputs. Although probing is typically not used as a stand alone approach, our preliminary experiment found that a vanilla probing baseline worked surprisingly well. However, we discover that current probe learning strategies are ineffective. We therefore propose Deep Linear **Probe Gen**erators (ProbeGen), a simple and effective modification to probing approaches. ProbeGen adds a shared generator module with a deep linear architecture, providing an inductive bias towards structured probes thus reducing overfitting. While simple, ProbeGen performs significantly better than the state-of-the-art and is very efficient, requiring between 30 to 1000 times fewer FLOPs than other top approaches.

## 1 INTRODUCTION

The growing importance and popularity of neural networks has led to the development of several model hubs (e.g. HuggingFace, CivitAI), where more than a million models are now publicly available. Treating them as a new data modality presents new opportunities for machine learning. Specifically, as not all neural networks include information about their training, developing methods to automatically learn from weights is becoming important. For instance, given an undocumented model, it is interesting to know its generalization error (Unterthiner et al., 2020) or its training dataset (Dupont et al., 2022). While some of these questions could be answered by evaluating the model on many labelled samples, this is often impractical as the data may be unavailable or unknown. Here, we want to answer these questions without access to the models true data distribution and under minimal computational effort.

Learning from weights is essentially similar to the well studied problem of binary code analysis Shoshitaishvili et al. (2016), where the goal is to predict the function of an unknown software using its binary code. In both cases, the task is to understand an unknown complex function specified by many parameters. Binary code analysis approaches generally fall into two categories: static and dynamic. Static methods Shoshitaishvili et al. (2016) aim to understand a function without running the binary code. Dynamic code analysis Egele et al. (2008); Bayer et al. (2006) runs the code on inputs provided by the user and analyzes its outputs to understand what the code does. Similarly, in weight space learning, there are two main types of methods: mechanistic approaches Unterthiner et al. (2020); Navon et al. (2023a); Zhou et al. (2024a); Lim et al. (2023); Kalogeropoulos et al. (2024) aim to understand model weights without running the model, while probing approaches Herrmann et al. (2024); Kofinas et al. (2024) represent models by their responses to a set of well selected inputs. Despite dynamic methods often performing better for binary code analysis, in the context of weight space learning, probing is still under utilized.

Motivated by the success of dynamic code analysis, in this paper we focus on advancing probing methods for learning from weights. First, we support this intuition by showing that a simple prob-

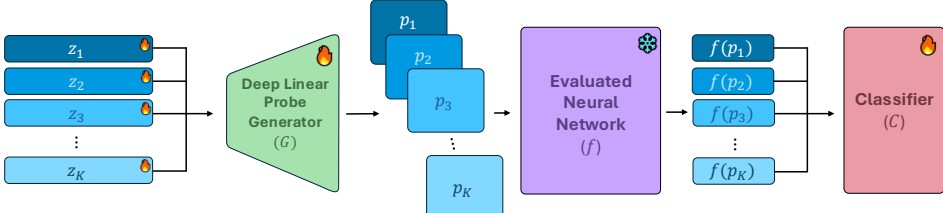

Figure 1: ***Overview of Our Method.*** We optimize a deep *linear* probe generator to create suitable probes for the model. Meaning, our generator includes no activations between its linear layers, yet the addition of linear layers reinforces a desired structure for the probes. We then gather the models responses over all probes, and train a classifier to predict some attribute of interest about the model.

ing baseline, with no bells-and-whistles, achieves comparable or better results than state-of-the-art mechanistic approaches. However, we discover that despite the good performance, current probing methods learn probes that perform comparably to random probes sampled from simple, unlearned statistical distributions. This suggests that current learned probes are suboptimal.

To this end, we propose *Deep Linear **Probe Gen**erators* (ProbeGen) as a simple and effective solution. ProbeGen factorizes its probes into two parts, a per-probe latent code and a global probe generator. The generator offers two key benefits: (i) It helps sharing information across multiple probes, and (ii) can implicitly introduce an inductive bias into the probes. For example, in images, hierarchical and convolutional layers create a local structure. Finally, by observing the learned probes we hypothesize they are not necessarily semantic, and owe some of their expression to low-level structures. We then find that the non-linear activation functions, which increase expressivity, actually degrade the learned probes. Our final approach therefore consists of a deep *linear* network (Arora et al., 2019), with data-dependent biases. Our linear generators produce probes that achieve state-of-the-art performance on common weight space learning tasks.

## 2 RELATED WORK

**Weight Space Learning.** A recent line of works have focused on training models to process the weights of diverse model populations (known as Model Zoos, e.g., (Schürholt et al., 2022)) to predict undocumented properties of a model. These properties include classifying the training dataset (e.g., the class of an image used to train an Implicit Neural Representation) or predicting the generalization error. Mechanistic methods (Lim et al., 2023; Navon et al., 2023a; Schürholt et al., 2024; Eilertsen et al., 2020; Unterthiner et al., 2020) represent models using the mechanics of their inner workings. One approach (De Luigi et al., 2023; Schürholt et al., 2024; 2021) is to learn standard architectures over raw weights. However, weights exhibit permutation symmetries between neurons (Navon et al., 2023a) which these architectures do not explicitly account for, although some of these methods use augmentations (Schürholt et al., 2024; 2021) to encourage permutation invariance. Other methods proposed specialized augmentations (Shamsian et al., 2024) which can be applied to aligned networks (Shamsian et al., 2024; Navon et al., 2023b; Peña et al., 2023; Ainsworth et al., 2022) or to already aligned weight spaces (Lim et al., 2024b;a). Another approach (Unterthiner et al., 2020; Dupont et al., 2022; Salama et al., 2024) maps the weights to a low dimensional embedding by a set of weight statistics, which are completely invariant to permutations, but have limited expressivity as they ignore the inner relations between neurons. Recently, a line of works (Navon et al., 2023a; Zhou et al., 2024a;b; Kofinas et al., 2024; Lim et al., 2023; Tran et al., 2024; Kalogeropoulos et al., 2024) focus on specially designing permutation equivariant architectures for processing neural networks. A dominant approach uses graph based architectures (Kofinas et al., 2024; Lim et al., 2023; Kalogeropoulos et al., 2024) modeling a neural network as a computational graph, where every neuron is a node. They then train Graph Neural Network (GNN) (Gilmer et al., 2017; Kipf & Welling, 2016) or Transformer (Vaswani, 2017; Diao & Loynd, 2022) modules, which are equivariant by design, to analyze the computational graph. Most recently, some equivariant approaches (Kofinas et al., 2024; Herrmann et al., 2024) also included learned probes. In this work we take a deep look into probing methods, and their failure points.

Another research thrust developed new applications for weight space learning. Some works (Ha et al., 2016; Ashkenazi et al., 2022; Peebles et al., 2022) encode the parameters of neural networks, mainly for generating, modifying or compressing weights matrices. Others (Erkoç et al., 2023; Dravid et al., 2024; Shah et al., 2023) use weights for advanced image generation capabilities. More recently, Carlini et al. (2024) proposed recovering the exact black-boxed weights of a neural network layer, and Horwitz et al. (2024a) demonstrated recovering entire pre-trained models when the weights of multiple fine-tuned models of the same Model Tree Horwitz et al. (2024b) are available.

**Implicit Neural Representations (INRs).** In recent years, INRs (Sitzmann et al., 2020; Tancik et al., 2020) have emerged as a new paradigm for representing data points with neural networks. These networks are used in various data modalities such as images (Ha, 2016), 3D shapes (Mescheder et al., 2019; Chen & Zhang, 2019), 3D scenes (Mildenhall et al., 2021), video (Li et al., 2021), audio (Sitzmann et al., 2020), etc. Specifically, in the context of images, INRs are neural networks trained on a single image, which given a pixel location $(x, y)$ return the value of that pixel in the training image. In this work, we classify the class of a training image an INR network was trained on, using only the INR's weights.

## 3 BACKGROUND

**Definition: Weight Classification.** The learner receives as input a set of $n$ models $f_1, f_2, \cdots, f_n$ where each has a corresponding label $y_1, y_2, \cdots, y_n$. Each model takes as input some tensor $x$ and outputs $f(x)$, where the output can be a vector of logits, probabilities or other variables. Each model is fully specified by its weights and architecture. The task of the learner is to train a classifier $C$, that takes as input the model $f$ and predicts its label $y$.

**The Challenge.** While a naive solution would be to apply a standard neural architecture directly to the weights, this idea encounters serious setbacks. The dimension of the weight vector is very high, but more fundamentally, the ordering of the neurons in each layer is arbitrary. Hence, different permutations of the neurons result in functionally identical models. Standard architectures such as MLPs and CNNs do not respect such symmetries. One popular approach by Unterthiner et al. (2020) computes permutation-invariant statistics for the flattened weights and biases of each layer, then trains a standard classifier on them. These statistics lose much of the information contained in the weights, limiting the potential of this approach. Another direction is learning with equivariant architectures (Navon et al., 2023a; Zhou et al., 2024a) which respect the permutation invariance, e.g., graph neural networks Kofinas et al. (2024); Lim et al. (2023). However, these methods (Kofinas et al., 2024) treat each neuron of the model as a token, and scaling them to large architectures is challenging (see Sec. 5.1).

**Probing.** Probing represents a model by running it on several fixed inputs and noting the responses received on them. The learner can then train a classifier to map the model responses to the label. This approach avoids the issue of weight permutation invariance as both the orders of input dimensions (e.g., image pixels) and output dimensions (e.g., class logits) are consistent across models.

Assuming that we want to predict an attribute $y$ of a network $f$, probing methods (Herrmann et al., 2024; Kofinas et al., 2024) optimize a set of $k$ probes $(p_1, ..., p_k)$, and feed them into the network. They then train a classifier $C$ on the concatenation of the outputs. The prediction $\hat{y}$ is:

$$\hat{y} = C(f(p_1), f(p_2), \cdots, f(p_k)) \tag{1}$$

Probing methods learn the parameters of each probe $p$ directly by latent optimization (Bojanowski et al., 2017). Each probe provides some information about the model attributes, and learning diverse and discriminative probes is key for obtaining a useful representation. The classifier $C$ leverages information from all probes, and is typically trained by cross-entropy for classification and mean squared error for regression.

## 4 WEIGHT SPACE LEARNING WITH DEEP LINEAR PROBE GENERATORS

Our initial hypothesis is that probing methods, when done right, hold significant potential. Much like binary code files, neural networks are unknown and highly complex functions. Drawing inspiration

Table 1: ***Simple Probing vs. Other Approaches.*** We compare a simple probing approach to previous graph based and mechanistic approaches. We average the results over 5 different seeds. For probing, we experiment with different numbers of probes (in brackets).

| | Accuracy | | Kendall's $\tau$ ($\uparrow$) | |
|---|---|---|---|---|
| Method | MNIST | FMNIST | CIFAR10-GS | CIFAR10 Wild Park |
| StatNN (0) | $0.398_{\pm 0.001}$ | $0.418_{\pm 0.002}$ | $0.914_{\pm 0.000}$ | $0.719_{\pm 0.010}$ |
| Neural Graphs (0) | $0.923_{\pm 0.003}$ | $0.727_{\pm 0.006}$ | $0.935_{\pm 0.000}$ | $0.817_{\pm 0.007}$ |
| Neural Graphs (64) | $0.967_{\pm 0.002}$ | $0.736_{\pm 0.012}$ | $0.938_{\pm 0.001}$ | $0.888_{\pm 0.009}$ |
| Neural Graphs (128) | $0.976_{\pm 0.001}$ | $0.745_{\pm 0.008}$ | $0.938_{\pm 0.000}$ | $0.885_{\pm 0.005}$ |
| Vanilla Probing (64) | $0.873_{\pm 0.026}$ | $0.784_{\pm 0.017}$ | $0.933_{\pm 0.001}$ | $0.885_{\pm 0.008}$ |
| Vanilla Probing (128) | $0.955_{\pm 0.005}$ | $0.808_{\pm 0.006}$ | $0.936_{\pm 0.001}$ | $0.889_{\pm 0.008}$ |

Figure 2: ***A Few Examples from the Dead Leaves Dataset.*** We show these images are synthetic and highly dissimilar to real images.

from binary code analysis, where dynamic approaches Egele et al. (2008); Bayer et al. (2006) are more common than static ones Shoshitaishvili et al. (2016), we believe that running neural networks, i.e., probing, is a promising approach for weight space learning. We begin with 2 preliminary experiments to test the quality and potential of probing approaches.

## 4.1 A SIMPLE PROBING BASELINE

As we believe probing should be an effective way of analyzing neural networks, we begin by testing its raw capabilities. We take a vanilla probing approach, without any enhancements or modifications, that optimizes all probes $p_1, p_2, ..., p_k$ with latent optimization (i.e., optimizing their values directly) and uses a simple MLP classifier for $C$. We compare this vanilla probing to its top competitors. First, a graph based approach (Neural Graphs) (Kofinas et al., 2024) which treats each neuron of the network as a node and operates a transformer on the resulting computational graph. Second, weight statistics (StatNN) (Unterthiner et al., 2020) which extracts simple statistics from the flattened weights and biases of a network and trains a simple predictor over them. We test all approaches on 4 popular benchmarks. For dataset classification, we measure the accuracy for MNIST LeCun et al. (1998) INRs digit prediction and Fashion-MNIST Xiao et al. (2017) INRs class prediction, both provided by Navon et al. (2023a). For generalization error prediction, we use the CIFAR10-GS (Unterthiner et al., 2020) and CIFAR10-Wild-Park (Kofinas et al., 2024) benchmarks, measuring the Kendall's $\tau$ metric as common in weight-space learning evaluation. The Kendall's $\tau$ is a statistic used to measure the correlation agreement between two rankings, where 1 indicates perfect correlation, and $-1$ indicates perfect negative correlation. The results are presented in Tab. 1. It is clear that (i) with enough probes, vanilla probing is able to perform better than a graph based approach that does not use probing. (ii) Graph based approaches become comparable to vanilla probing only when incorporating probing features. This shows the promise of probing methods. Additionally, in Sec. 5.1 we demonstrate that probing also requires much less computational resources than graph based methods.

## 4.2 LEARNED VS. UNLEARNED PROBES

Having shown the merit in probing approaches, it is interesting to understand the quality of the probes themselves. To do so, we replace the learned probes by a set of unlearned inputs, training only the classifier $C$. We test both: (i) probes with no knowledge about the training data, selected from random synthetic data, and (ii) probes from in-distribution data, selected from the training set of the networks. We fix the number of probes $k$ in all cases. In the MNIST and FMNIST INRs

Table 2: ***Learned vs. Out of Distribution Probes.*** Comparison of learned probes, in-distribution data probes and probes from randomly selected data. We average the results over 5 different seeds.

| # Probes | Method | Accuracy | | Kendall's $\tau$ ($\uparrow$) | |
| | | MNIST | FMNIST | CIFAR10-GS | CIFAR10 Wild Park |
|---|---|---|---|---|---|
| 64 | Learned Probes | 0.873 $\pm0.026$ | 0.784 $\pm0.017$ | 0.933 $\pm0.001$ | 0.885 $\pm0.008$ |
| | Synthetic Probes | 0.899 $\pm0.022$ | 0.832 $\pm0.010$ | 0.918 $\pm0.001$ | 0.882 $\pm0.007$ |
| | In-Dist. Probes | 0.899 $\pm0.022$ | 0.832 $\pm0.010$ | 0.939 $\pm0.000$ | 0.922 $\pm0.008$ |
| 128 | Learned Probes | 0.955 $\pm0.005$ | 0.808 $\pm0.006$ | 0.936 $\pm0.001$ | 0.889 $\pm0.008$ |
| | Synthetic Probes | **0.959** $\pm0.006$ | **0.856** $\pm0.007$ | 0.917 $\pm0.001$ | 0.893 $\pm0.014$ |
| | In-Dist. Probes | **0.959** $\pm0.006$ | **0.856** $\pm0.007$ | **0.941** $\pm0.001$ | **0.930** $\pm0.011$ |

Figure 3: ***Learned Probes.*** Optimized probes by (a) Latent Optimization and (b) ProbeGen for the CIFAR10 Wild Park benchmark. We show the same probes for identical runs (including the seed), except the generator module.

tasks, synthetic and in-distribution probes are the same, where we choose points from a uniform distribution between $[[-1, 1], [-1, 1]]$. In the CIFAR10 cases, we select $k$ random synthetic images from the Dead Leaves (Baradad Jurjo et al., 2021; Lee et al., 2001) dataset as synthetic probes, and $k$ random images from CIFAR10 as in-distribution probes. As seen in Fig. 2, Dead Leaves images are synthetic, unlearned and highly dissimilar to real images. We therefore choose these images as our synthetic probes as they include some structure, yet are still far from being realistic. The results are presented in Tab. 2. It is clear that random probes are comparable to learned ones. We conclude that current probing techniques find suboptimal probes. To understand why these learned probes perform worse, we first observe them in Fig. 3a. The probes show low-level, almost adversarial patterns, which can be highly expressive (Goodfellow et al., 2014). We therefore hypothesize vanilla probing tends to overfit. In Sec. 5.2 we demonstrate that our final method indeed overfits less than vanilla probing, and that more expressive methods hurt performance in these tasks.

## 4.3 DEEP LINEAR PROBE GENERATORS

We propose Deep Linear **Probe Gen**erators (ProbeGen) for learning better probes. ProbeGen optimizes a deep generator module limited to linear expressivity, that shares information between the different probes. It then observes the responses from all probes, and trains an MLP classifier on them. While simple, in Sec. 5.1 we show it greatly enhances probing methods, and also outperforms other approaches by a large margin.

**Shared Deep Generator.** Learning the probes through latent optimization prevents them from sharing useful patterns, as they do not have any shared parameters. A straightforward way for overcoming this is by factorizing each probe $p_i$ into two parts: (i) a latent code $z_i$ learned using latent optimization and (ii) a deep generator network $G$ that all probes share. Formally,

$$p_i = G(z_i) \tag{2}$$

where the latent code $z_i$ can have either a higher or a lower dimension than $p_i$. This factorization introduces a dependence between the probes as they all share $G$. It also reduces their expressivity, as $G$ may not be able to express all possible outputs.

**Deep Linear Networks.** Non-linear activations such as ReLU are the engine that makes deep networks very expressive. However, in this case, we would like to add regularization (see Sec. 4.2) rather than increasing expressivity. We therefore remove the activations from our generator, keeping only stacked linear layers. Work by Arora et al. (2019) showed that when using SGD, deep linear networks have an implicit regularization effect. We therefore use deep linear networks in our approach, i.e., stacked linear layers but without the non-linear activations between them. In Sec. 5.2, we show that removing the activations reduces overfitting and therefore also enhances performance.

**Data-Specfic Inductive Bias.** In the case that our target model $f$ takes structured inputs such as images, we hypothesize that presenting probes from a similar distribution will achieve higher accuracy and reduce overfitting. As we do not know the data distribution, we cannot learn an accurate generative model for it. Still, we can introduce data-specific inductive biases into the generator. For example, we present a generator for images, where we simply stack 2D convolutional layers, similarly to DCGAN (Radford et al., 2015). While this does not guarantee natural image statistics, it at least encourages some local structure and multi-scale hierarchy, both of which are some of the most important image characteristics.

## 5 EXPERIMENTS

Our evaluation follows the standard protocol for weight space learning. We evaluate on two tasks: (i) CNNs generalization error prediction and (ii) detecting the training classes of images based on INR networks trained on them. We include experiments on small-scale established benchmarks as well as a new larger-scale Model Zoo which we present, using ResNet18(He et al., 2016) models.

**Baselines.** We compare our method, ProbeGen, with StatNN (Unterthiner et al., 2020), DWS(Navon et al., 2023a), NFN (Zhou et al., 2024a), ScaleGMN (Kalogeropoulos et al., 2024) and Neural Graphs (Kofinas et al., 2024). StatNN computes 7 statistics for the weights and biases of each layer, concatenates them and trains a gradient boosted tree method on this representation. DWS (Navon et al., 2023a) and NFN (Zhou et al., 2024a) train linear, permutation equivariant layers with non-linearities between them, but DWS is not applicable for CNNs. ScaleGMN (Kalogeropoulos et al., 2024) treats neural networks as computational graphs, accounting for permutation symmetries and scaling symmetries. Neural Graphs (Kofinas et al., 2024) is another graph-based approach, where each bias is a node and the weights are the matching edges between these nodes. This method then trains a transformer on the created graph, so the attention score between a pair of neurons depends on the weight that connects them. While Neural-Graphs do not account for scaling symmetries like the ScaleGMN baseline, it enriches each neuron's representation by using probing features.

**Datasets.** We evaluate on 4 established datasets. For training data prediction we choose the MNIST and FMNIST implicit neural representation (INR) benchmarks (Navon et al., 2023a). Both datasets were formed by training an INR Sitzmann et al. (2020) model for each image of the original dataset. The goal is predicting the class of an image given its INR network. For generalization error prediction, we used the CIFAR10-GS (Unterthiner et al., 2020) and CIFAR10 Wild Park tasks Kofinas et al. (2024). These datasets consists of thousands of small CNN models ($3-5$ layers), each trained separately on CIFAR10 (Krizhevsky et al., 2009). We use accuracy for weight classification and Kendall's $\tau$ for regression.

**Metrics.** For INR class prediction, we use simple accuracy to measure performance. For predicting the generalization error of CNNs, we follow standard evaluation protocol and use the Kendall's $\tau$ metric, which measures the agreement between two rankings. Similar to Pearson's correlation, with Kendall's $\tau$, values near 1 indicate strong positive correlation, values near $-1$ indicate strong negative correlation, and values near 0 indicate no correlation.

Table 3: ***Results for Small Scale Benchmarks.*** Comparison of ProbeGen, to graph based, mechanistic approaches and latent optimized probes. We average the results over $5$ different seeds.

| # Probes | Method | Accuracy | | Kendall's $\tau$ ($\uparrow$) | |
| --- | --- | --- | --- | --- | --- |
| | | MNIST | FMNIST | CIFAR10-GS | CIFAR10 Wild Park |
| | StatNN | $0.398$ $_{\pm 0.001}$ | $0.418$ $_{\pm 0.002}$ | $0.914$ $_{\pm 0.000}$ | $0.719$ $_{\pm 0.010}$ |
| | DWS | $0.857$ $_{\pm 0.006}$ | $0.671$ $_{\pm 0.003}$ | - | - |
| 0 | $NFN_{HNP}$ | $0.791$ $_{\pm 0.008}$ | $0.689$ $_{\pm 0.006}$ | $0.934$ $_{\pm 0.001}$ | - |
| | $ScaleGMN_B$ | $0.966$ $_{\pm 0.002}$ | $0.808$ $_{\pm 0.001}$ | $0.941$ $_{\pm 0.000}$ | - |
| | Neural Graphs | $0.923$ $_{\pm 0.003}$ | $0.727$ $_{\pm 0.006}$ | $0.935$ $_{\pm 0.000}$ | $0.817$ $_{\pm 0.007}$ |
| | Neural Graphs | $0.967$ $_{\pm 0.002}$ | $0.736$ $_{\pm 0.012}$ | $0.938$ $_{\pm 0.001}$ | $0.888$ $_{\pm 0.009}$ |
| 64 | Vanilla Probing | $0.873$ $_{\pm 0.026}$ | $0.784$ $_{\pm 0.017}$ | $0.933$ $_{\pm 0.001}$ | $0.885$ $_{\pm 0.008}$ |
| | **ProbeGen** | $0.980$ $_{\pm 0.001}$ | $0.861$ $_{\pm 0.004}$ | $0.956$ $_{\pm 0.000}$ | **$0.933$** $_{\pm 0.005}$ |
| | Neural Graphs | $0.976$ $_{\pm 0.001}$ | $0.745$ $_{\pm 0.008}$ | $0.938$ $_{\pm 0.000}$ | $0.885$ $_{\pm 0.005}$ |
| 128 | Vanilla Probing | $0.955$ $_{\pm 0.005}$ | $0.808$ $_{\pm 0.006}$ | $0.936$ $_{\pm 0.001}$ | $0.889$ $_{\pm 0.008}$ |
| | **ProbeGen** | **$0.984$** $_{\pm 0.001}$ | **$0.877$** $_{\pm 0.003}$ | **$0.957$** $_{\pm 0.001}$ | $0.932$ $_{\pm 0.006}$ |

Table 4: ***FLOPs Comparison.*** Using $128$ probes and a batch size of $64$. ProbeGen is much more efficient than graph approaches.

Table 5: ***Results for ResNet Scale.*** ProbeGen can successfully scale to larger sized architectures, and outperforms both baselines.

| Method | Billion FLOPs ($\downarrow$) | |
| --- | --- | --- |
| | MNIST | CIFAR10-GS |
| Neural Graphs | 63.40 | 94.56 |
| **ProbeGen** | **0.02** | **3.41** |

| # Probes | Method | Kendall's $\tau$ ($\uparrow$) |
| --- | --- | --- |
| | | ResNet18 Zoo |
| 0 | StatNN | 0.856 |
| 128 | Vanilla Probing | 0.843 |
| 128 | **ProbeGen** | **0.910** |

## 5.1 MAIN RESULTS

**Small-Scale Performance.** Tab. 3 summarizes the results on the standard benchmarks. On INR dataset prediction tasks, ProbeGen outperforms all benchmarks significantly, even when the baselines use the same number of probes. E.g., in the FMNIST class prediction task, ProbeGen outperforms all other approaches by more than $6\%$, reaching an accuracy of $87.7\%$. ProbeGen also outperforms the baselines on predicting the generalization error of CNNs. While vanilla probing performs similarly to graph approaches on CIFAR10-GS, ProbeGen is clearly able to improve the quuality of the probes, and achieves the highest result. On CIFAR10 Wild Park ProbeGen also outperforms all previous approaches significantly, improving Kendall's $\tau$ from $0.889$ to $0.933$.

**Computational Cost.** We compare the computational cost of ProbeGen and Neural-Graphs in terms of floating point operations (FLOPs). We test on the MNIST INRs and CIFAR10-GS datasets, with $128$ probes and a batch size of $64$ for both approaches in Tab. 4. We see that probing is a much more efficient approach, requiring between $1.5 - 3$ orders of magnitude fewer FLOPs.

**Large models.** Knowing that probing methods are more efficient, we now wish to test if probing methods can really scale to larger size models. We create a new dataset of over $6,000$ ResNet18 He et al. (2016) models. Each ResNet model was trained on a a randomly selected subset of Tiny-Imagenet Le & Yang (2015); Deng et al. (2009). We sampled the subset out of a closed list of $10$ subsets, that we created in advance. For each model, we record its generalization (test) error, and the objective is to predict this error given the model weights. Graph based methods were too computationally expensive in this scale. Tab. 5 shows the results. ProbGen achieves the best results reaching a $0.91$ kendall's $\tau$, significantly outperforming both vanilla probing and statNN.

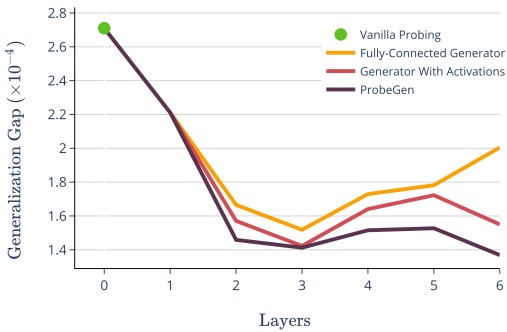
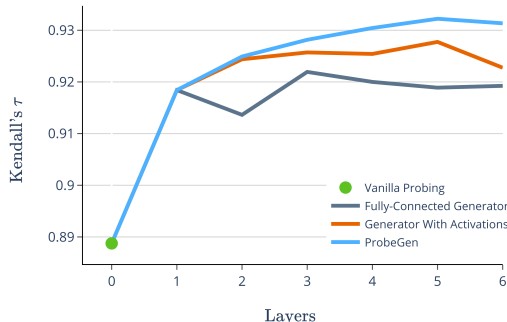

Figure 4: ***Overfitting for Different Generators.*** We compare ProbeGen to linear generators with fully connected layers, and non-linear convolutional generators. Overfit is measured by the generalization gap of each method. Results are averaged over 5 seeds.

Figure 5: ***Ablation Studies.*** We compare the performance of ProbeGen, linear generators with fully connected layers, and non-linear convolutional generators, for varying numbers of layers. With less than 2, all versions are equivalent to ProbeGen. We average over 5 seeds.

Table 6: ***Number of Probes***. We compare ProbeGen to vanilla probing with different numbers of learned probes. We average over 5 different seeds.

| # Probes | FMNIST (↑) | | CIFAR10 Wild Park (↑) | |
| --- | --- | --- | --- | --- |
| | Vanilla | ProbeGen | Vanilla | ProbeGen |
| 16 | $0.686_{\pm0.043}$ | $0.805_{\pm0.006}$ | $0.882_{\pm0.010}$ | $0.926_{\pm0.007}$ |
| 32 | $0.764_{\pm0.013}$ | $0.844_{\pm0.006}$ | $0.884_{\pm0.008}$ | $0.930_{\pm0.005}$ |
| 64 | $0.784_{\pm0.017}$ | $0.861_{\pm0.004}$ | $0.885_{\pm0.008}$ | $0.933_{\pm0.005}$ |
| 128 | $0.808_{\pm0.006}$ | $0.877_{\pm0.003}$ | $0.889_{\pm0.008}$ | $0.932_{\pm0.006}$ |

## 5.2 ABLATION STUDIES

**Linear Generators.** By observing Fig. 3, we hypothesized that removing the activations between the linear layers of the generator will reduce its overfitting. We therefore compare the results with and without non-linear activations. Indeed, as seen in Fig. 4, using non-linear activations results in a higher generalization gap, i.e., more overfitting. The amount of overfitting is even worse when not using a generator at all. In Fig. 5 we see that these activations also harm the model's performance, suggesting they lead to probes with reduced generalization abilities. In App. A we provide ablations demonstrating that, for these tasks, a deep linear generator outperforms other regularizations.

**Structure of Probes.** First, we show several probes optimized via latent optimization and the same ones when optimized by ProbeGen (see Fig. 3). Although both not interpetable by humans, it is clear that ProbeGen probes have much more structure than latent-optimized ones. In Fig. 4 we demonstrate this has a regularizing effect, as ProbeGen significantly reduces the generalization gap compared to vanilla probing.

Additionally, we also test the contribution of the local inductive bias. By replacing the convolutional layers in our generator to fully-connected ones, we completely remove this bias from the generator. Comparing the fully connected generator to our original ProbeGen helps isolate the effect of the local bias, as: (i) Both versions are linear, meaning the fully-connected model is less restricted, and (ii) the size of the feature maps is kept similar throughout the generation process in both versions. Therefore, the primary difference is the inductive bias introduced by the convolutional layers. We compare the results in Fig. 5 for different numbers of layers. Indeed, we see the generators inductive bias is important, as ProbeGen consistently outperforms the fully connected version.

**Number of Probes.** We compare ProbeGen to vanilla probing for differing numbers of probes. The results in Tab. 6, show that ProbeGen significantly improves over vanilla probing, even when using only a fraction of the number of probes. In the FMNIST case, ProbeGen with 32 probes

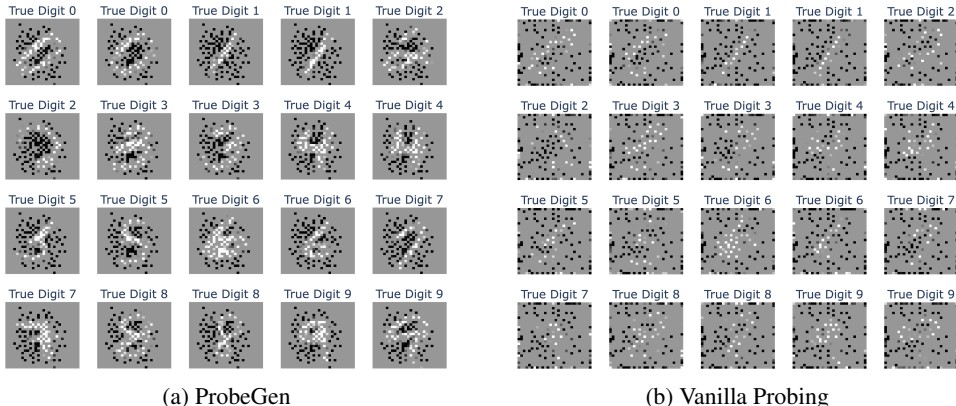

(a) ProbeGen         (b) Vanilla Probing

Figure 6: **MNIST Representations Extracted by ProbeGen vs. Vanilla Probing.** We place the INR prediction in the probe locations. Other pixels are in gray.

already surpasses vanilla probing with $128$ probes by almost $4\%$ accuracy. In the CIFAR10 case 16 ProbeGen probes are already significantly better than $128$ latent optimized ones.

**Representation Interpretability.** ProbeGen represents each model as an ordered list of output values based on carefully chosen probes. These representations often have semantic meanings as the output space of the model (here, image pixels or logits) are semantic by design. For the MNIST INRs dataset, we visualize the inputs and outputs together, showing the partial image created by the probes. Fig. 6 displays several representations for a few randomly selected images, comparing ProbeGen with vanilla probing. Vanilla probing chooses locations scattered around the image, including pixels far out of the image, where the behaviour of the INR is unexpected. ProbeGen on the other hand, chooses object centric locations, as suitable for this task. Indeed, one can easily identify the digits in the images despite only observing less than 20% of their pixels, hinting that this probe selection simplifies the task for the classifier module.

In Fig. 7 we visualize the representation (i.e., logits) of ProbeGen when trained on the CIFAR10 Wild Park dataset. We can see that the values become more uniform as the accuracy of the models decreases, and sharper as it increases. This suggests that ProbeGen uses some form of prediction entropy in its classifier. We further test this hypothesis by training a classifier that only takes the entropy of each probe as its features. We find that while not as effective as ProbeGen, this classifier is still able to achieve a Kendall's $\tau$ of $0.877$. Hence, even when taken alone, prediction entropy is in fact highly discriminative for this task.

## 6   DISCUSSION

**Black-box settings.** We showed the potential of probing for learning from models. A side benefit of probing, compared to other approaches, is that it is suitable for inference on black-box models without any further adjustments. As probing only observes models responses, at inference time it could simply probe a black-box model by its API, then make a prediction based on the outputs.

**Other modalities.** ProbeGen incorporates inductive biases into the probe generator module. We tested it on images, showing its potential when having the right inductive bias, however, other modalities could require different structural biases. E.g., in audio, a consistency term would probably be helpful to simulate realistic recordings, and textual data may even require some pre-training for linguistic priors of the probes. In App. B we present a preliminary experiment using INRs trained on point clouds, demonstrating that ProbeGen's success generalizes to other modalities as well.

**Adaptive probing.** One interesting future direction for improving probing is being able to adaptively choose the probes used to test each model. Probing models can than learn a policy for this adaptive selection, which would potentially improve accuracy for a given certain probe budget. An early work by Herrmann et al. (2024) showed this could be effective for sequential models.

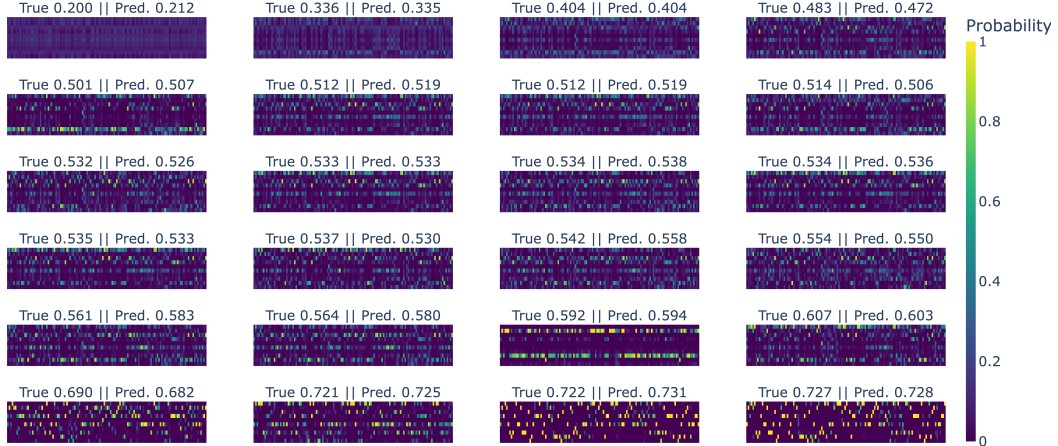

Figure 7: **CIFAR10 Wild Park Representations Extracted by ProbeGen's Learned Probes.** Each representation includes the 10 predicted probabilities (rows) given ProbeGen's 128 probes (columns), and above it are its true and predicted accuracies. Samples are sorted by true accuracy.

## 7    LIMITATIONS

**Output space structure.**    While we demonstrated probing is a powerful tool for learning from neural networks, it requires the input and output dimensions to retain the same meaning across models. There are important cases that do not satisfy this requirement, e.g., a repository of classification models with different classes in each model. Here, the output space for each varies and even if models share the same classes their order may still differ. Extending probing method to deal with the above cases is an important direction for future work.

**Weight generative tasks.**    Probing only looks at the input and output layers of each mode. Therefore, it cannot be used to give layer or weight level predictions. That means that it is not suitable for weight generation tasks, such as editing or creating new neural networks.

**Scalability.**    In Sec. 5.1 we showed probing is much more computationally efficient than previous graph based approaches. Still, it requires forwarding the entire model a few times and computing the gradients through the model, with multiple models in each batch. This means that in order to infer about a model using probing, one would need computational resources equivalent to training such a model. This would require a non-trivial solution for learning from larger models, e.g. CLIP (Radford et al., 2021) or Stable Diffusion (Rombach et al., 2022), under a limited compute budget.

## 8    CONCLUSIONS

This paper championed probing methods for weight space learning and improved them to achieve better than state-of-the-art performance. We first showed that a vanilla probing approach, based on latent optimization, outperforms previous methods. However, we found that the learned probes are no better than randomly sampled synthetic data. To learn better probes, we proposed deep linear generator networks that significantly reduce overfitting through a combination of implicit regularization and data-specific inductive bias. Beside consistently achieving the highest performance, often by a large margin, our method requires 30 to 1000 fewer FLOPs than other top methods.

## 9    SOCIAL IMPACT

The goal of this paper is to further our understanding of neural networks. We expect this increased understanding to be helpful for reducing the social risks of AI such as bias and model safety.

## 10 ACKNOWLEDGMENTS

This work was partially supported by the Israel Science Foundation (ISF), the Council for Higher Education (Vatat), the Center for Interdisciplinary Data Science Research (CIDR), the Israeli Cyber Authority, and KLA.

## 11 REPRODUCABILLITY

In this work, we presented a new and light-weight framework for weight space learning. Our method is simple to implement, and can be easily reproduced. To encourage future work in this direction, we provide a short implementation of our method in the supplementary materials.

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

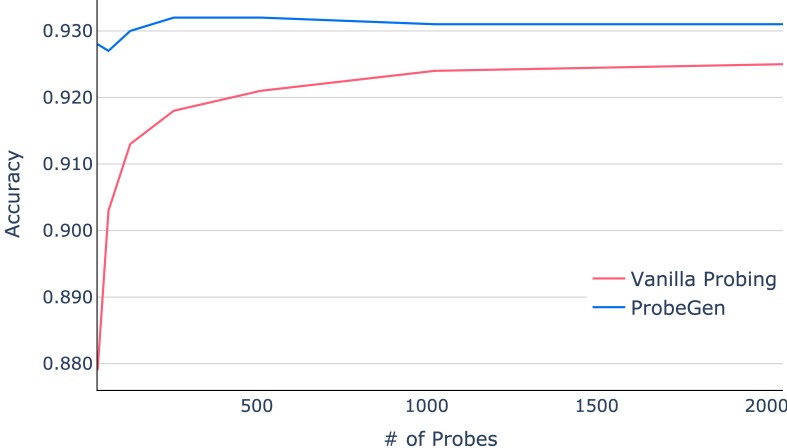

Figure 8: ***Ablating the Number of Probes used for Point Cloud INRs Classification.*** We show that ProbeGen exhibits a much better scaling law when classifying point clouds, compared to Vanilla Probing. Notably, ProbeGen with 32 probes already outperforms Vanilla Probing with 2048 probes.

## A   OTHER REGULARIZATIONS

We compare deep linear generators to a range of $\ell_2$ weight decay regularization strengths. We present the results in Tab. 7. Our results show that while adding some regularization might be helpful, it is not as good as a deep linear generator, and requires aggressive hyper-parameter tuning of the regularization strength, including non-standard values (a weight decay of $10^{-7}, 10^{-8}$), where a deep linear generator does not. Moreover, these regularizations are not consistent between experiments, and there is no single value that fits all experiments. We conclude that a deep linear generator, as presented in ProbeGen, provides a stable and subtle regularization that fits weight-space analysis.

Table 7: ***Performance Comparison of Different Regularizations vs. ProbeGen.***

|  | Accuracy | | Kendall's $\tau$ ($\uparrow$) | |
| Method | MNIST | FMNIST | CIFAR10-GS | CIFAR10 Wild Park |
| --- | --- | --- | --- | --- |
| Non-Linear Gen. + L2 Reg. ($10^{-2}$) | 0.942 | 0.531 | 0.758 | 0.442 |
| Non-Linear Gen. + L2 Reg. ($10^{-3}$) | 0.979 | 0.858 | 0.824 | 0.547 |
| Non-Linear Gen. + L2 Reg. ($10^{-4}$) | 0.981 | 0.867 | 0.861 | 0.737 |
| Non-Linear Gen. + L2 Reg. ($10^{-5}$) | 0.982 | 0.869 | 0.911 | 0.871 |
| Non-Linear Gen. + L2 Reg. ($10^{-6}$) | 0.982 | 0.870 | 0.941 | 0.905 |
| Non-Linear Gen. + L2 Reg. ($10^{-7}$) | 0.982 | 0.869 | 0.947 | 0.922 |
| Non-Linear Gen. + L2 Reg. ($10^{-8}$) | 0.981 | 0.870 | 0.948 | 0.928 |
| ProbeGen | 0.984 | 0.877 | 0.957 | 0.932 |

## B   ANOTHER DATA MODALITY: POINT CLOUDS

We extend our evaluation to include a more complex data modality: point clouds. Specifically, we use the Neural-Field-Arena (Papa et al., 2024), to evaluate ProbeGen's ability in classifying INRs trained on point clouds from the ShapeNet (Chang et al., 2015) dataset. The goal is to detect the class of the point cloud each INR was trained on, directly from the INR's weights. As shown in Table 8, ProbeGen consistently outperforms all prior approaches, including Vanilla Probing.

Figure 8 further examines the efficiency of probing methods with respect to the number of probes used. We can see that even under a highly restrictive budget of 32 probes, ProbeGen is able to surpass all baselines. Notably, this 32 probes version of ProbeGen outperforms Vanilla Probing with 2048 probes, showing ProbeGen's better scaling laws.

Table 8: ***Point Cloud INRs Classification.*** We show that ProbeGen can successfully learn from models trained on more complex data modelities, such as point clouds. Number of probes is in parentheses.

| Method | Accuracy
Point Clouds INRs |
|---|---|
| DWS | 0.911 |
| Neural Graphs | 0.903 |
| Vanilla Probing (128) | 0.913 |
| **ProbeGen** | **0.930** |

## C  IMPLEMENTATION DETAILS

**Generator Architecture.**    For image generation, we use a transposed convolution based generator. With each layer, the feature maps spatial sizes doubles in each axis (and 4 times overall), while the number of neurons (channels) decrease by half. We choose a generator width multiplier of 16, i.e., our generators last layer has 16 input channels, and the channels multiply by 2 with each previous layer. Our image generators has 6 transposed convolutional operators for experiments on the CIFAR10-GS and ResNet18 cases, and 5 for experiments on the CIFAR10 Wild Park dataset.

For our INR coordinates generators we use Fully-connected layers, with a hidden size of 32. These generators use 2 fully-connected layers for both FMNIST INRs and MNIST INRs.

**Hyper-parameters.**    We use a learning rate of $3 \cdot 10^{-4}$ and a batch size of 32 in all our experiments. Our MLP classifier $C$, uses 6 layers with a hidden size of 256. The latent vectors of each probe are of size 32. We trained all probing algorithms on the INR and CIFAR10 Wild Park experiments for 30 epochs, all experiments on the CIFAR10-GS dataset for 150 epochs, and all experiments on our ResNet18 Model Zoo dataset for 100 epochs. Additionally, we trained all baselines using their default hyper-parameters.

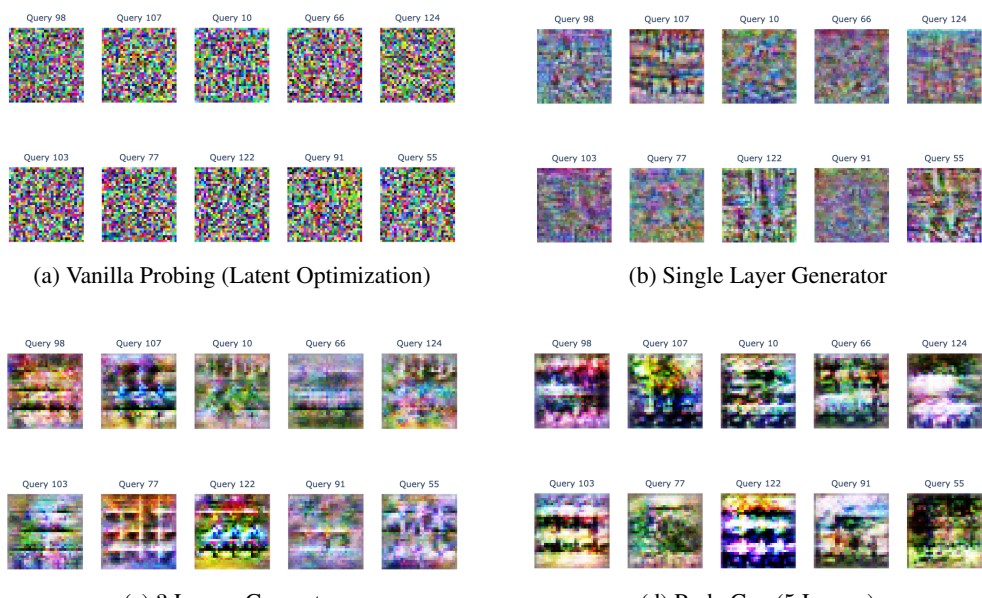

(a) Vanilla Probing (Latent Optimization)

(b) Single Layer Generator

(c) 3 Layers Generator

(d) ProbeGen (5 Layers)

Figure 9: ***Learned Queries by Deep Linear Generators.*** Queries learned by deep linear generators and vanilla probing, for the CIFAR10 Wild Park dataset.

**Fully-Connected Image Generators.** We find that optimizing fully connected generators with the exact same dimensions as ProbeGen leads to sever overfitting. Therefore, instead of having $C \times H \times W$ hidden dimensions (where $H$ and $W$ are the spatial sizes created by the convolutions) in each layer, we choose $3 \times H \times W$ which empirically worked much better.

# D QUERIES OF DIFFERENT PROBING APPROACHES

We qualitatively compare the learned queries by different probing algorithms. We visualize the same random subset of the queries from each algorithm, learned for the CIFAR10 Wild Park dataset.

In Fig. 9, we provide queries learned by our ProbeGen using different numbers of layers. We see the queries gradually develop structure as the number of layers increases. This goes in line with our hypothesis from Sec. 4.3.

Next, we provide a visualization of the queries of ProbeGen with activations between its linear layers. These queries are provided in Fig. 10. We see a more repetitive structure in these queries than in the standard ProbeGen, indicating the information may reside to more local patterns, which are not necessarily object centric as CIFAR10 tends to be.

Finally, we observe the queries from a fully-connected generator. Presented in Fig. 11, these queries show very little structure even when the generator have 5 layers. The structure is of local patterns, as there are no convolutions to present a hierarchical order. This shows the convolutional operators indeed enforce the desired structure on its queries.

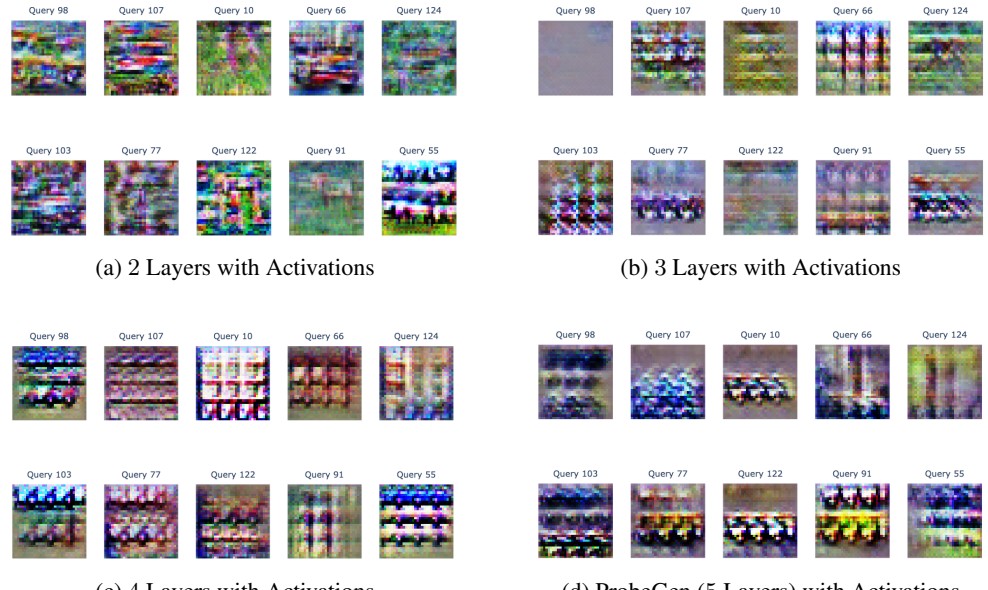

Figure 10: ***Learned Queries by Non-Linear Generators.*** Queries learned by non-linear generators for the CIFAR10 Wild Park dataset.

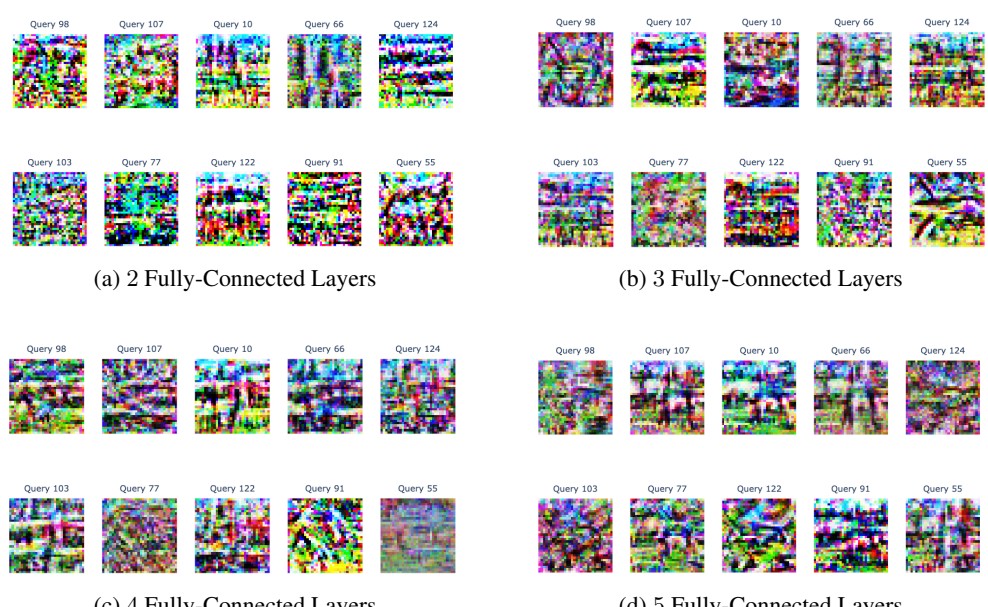

(a) 2 Fully-Connected Layers

(b) 3 Fully-Connected Layers

(c) 4 Fully-Connected Layers

(d) 5 Fully-Connected Layers

Figure 11: *Learned Queries by Fully-Connected Linear Generators.* Queries learned by fully-connected linear generators for the CIFAR10 Wild Park dataset.

