# OpenReview forum: "Deep Linear Probe Generators for Weight Space Learning"
_ICLR.cc/2025/Conference — ICLR 2025 Poster_

### Official Review · Reviewer_DBUB · 2024-10-26

**Soundness:** 2
**Presentation:** 3
**Contribution:** 2
**Rating:** 6
**Confidence:** 4

**Summary:**

The paper considers the problem of weight space learning, in which a learner is tasked with predicting the properties (e.g., accuracy) of an input (trained) neural net. The paper makes the distinction between two approaches for weight space learning: (1) Mechanistic approaches, which operate on the input weights (without running the model), and (2) Probing approaches, which represent models by their responses to a set of inputs. The paper champions probing approaches and proposes a simple yet effective approach in which the probes (model inputs) are learned using a deep linear generator. On a set of small-scale benchmarks, the authors show the proposed method outperforms both probe-based and mechanistic baselines, with improved computational efficiency.

**Strengths:**

- The paper is well-structured and easy to follow.
- The method is well motivated through a set of empirical observations.
- The proposed approach is simple yet effective.

**Weaknesses:**

My main concern lies with the experimental setup, which is limited to small-scale datasets and includes only two baselines. In addition, all experiments are performed on images with no additional modalities. This restricts the generalizability of the findings. I would expect a broader and more diverse set of experiments (in terms of scale and modalities) from a purely empirical paper. Please consider adding additional experiments with larger models, higher-dimensional inputs, or different modalities.

Additional limitations:
1. The approach is not applicable for an important and large set of weight space tasks, namely weights-to-weights tasks, e.g., INR editing, domain adaptation, sparsification, etc.
1. The provided empirical evaluation is limited to fairly low-dimensional input spaces. I assume probing approaches will be more efficient in low-dim spaces, so it will be beneficial to provide results for more challenging setups with higher-dimensional input spaces (e.g., point clouds) or input spaces with more challenging structures (e.g., graphs).
1. Similarly, results for additional data modalities will be beneficial and will strengthen the empirical soundness.
1. Also, including additional baselines like [1, 2] would be beneficial.
1. While the authors analyze the computational cost, showing clear benefits compared to the GNN-based approach, it is not clear that the results will hold for larger models, as the approach requires forward and backward steps through the models.
1. The Appendix is completely missing, along with additional important information regarding implementation details, experimental setup, HPs, etc.

References:

[1] Kalogeropoulos et al. "Scale Equivariant Graph Metanetworks," NeurIPS 2024.

[2] Schürholt et al. "Towards Scalable and Versatile Weight Space Learning." ICML 2024.

**Questions:**

- Have you tried comparing to using real images from the training dataset (for example, CIFAR10 images) as probes?
- It is not clear if you also used Conv layers for the INR tasks. If so, why does this make sense?

---

> ### Author Response · Authors · 2024-11-20
> **Response (1/2) to Reviewer DBUB**
>
> We thank the reviewer for their effort and for finding our method to be “simple”, “effective” and “well motivated”. Below, we address the reviewer’s concerns in detail.
>
> ___
>
> > *My main concern lies with the experimental setup, which is limited to small-scale datasets...*
>
> We conducted a larger scale experiment on ResNet18. The largest Model Zoo originally evaluated in our paper, contains models with 17.5K parameters on average. **In contrast, our ResNet18 models have over 11.1M parameters, more than a 600x scale up.** We trained over $6,000$ ResNet18 models using different subsets of Tiny-ImageNet, where the weight learning task is to predict their test accuracy. Each subset is of $10$ classes, and there are $10$ different subsets in total.
>
> The results are presented in the table below. ProbeGen scaled to larger models, reaching a kendall's $\tau$ of $0.91$ compared to statNN and Vanilla Probing which achieve $0.86$ and $0.843$ respectively. Neural-Graphs were too computationally expensive to scale to this experiment. We will release this new Model Zoo to encourage further research in this area.
>
> | Method           | Kendall's $\tau$          |
> |------------------|---------------------------|
> | StatNN           |          0.860            |
> | Neural Graphs    |            -              |
> | Vanilla Probing  |          0.843            |
> | ProbeGen         |          **0.910**        |
>
> This experiment demonstrates that ProbeGen can effectively scale to models with millions of parameters.
>
> ___
>
> > *Also, including additional baselines like [1, 2] would be beneficial.*
>
> We added several additional baselines including [1], to our main experimental results. We provide the modified tables below. It was not trivial for us to adapt [2] to run on the current datasets and could not include it. We also modified these in the revised manuscript, where changes are marked in yellow.
>
> INR Results:
> | Method              | MNIST  | FMNIST |
> |---------------------|--------|--------|
> | StatNN              | 0.398  | 0.418  |
> | DWS                 | 0.857  | 0.671  |
> | NFN_{HNP}          | 0.791  | 0.689  |
> | ScaleGMN_{B}       | 0.966  | 0.808  |
> | Neural-Graphs  | 0.976  | 0.745  |
> | Vanilla Probing     | 0.955  | 0.808  |
> | ProbeGen            | 0.984  | 0.877  |
>
>
> CNN Results:
>
> | Method              | CIFAR10-GS | CIFAR10 Wild Park |
> |---------------------|------------|--------------------|
> | StatNN              | 0.914      | 0.719             |
> | NFN_{HNP}          | 0.934      | -                 |
> | ScaleGMN_{B}       | 0.941      | -                 |
> | Neural-Graphs  | 0.938      | 0.885             |
> | Vanilla Probing     | 0.936      | 0.889             |
> | ProbeGen            | 0.957      | 0.932             |
>
> ___
>
> > *The approach is not applicable for an important and large set of weight space tasks, namely weights-to-weights tasks, e.g., INR editing, domain adaptation, sparsification, etc.*
>
> Indeed, our approach focuses on discriminative rather than generative weight-space tasks. However, we stress that discriminative tasks are important, have a very wide scope, and we barely scratched the surface. For instance, fields like membership inference attacks and model calibration aim to predict a network’s accuracy for specific samples, and are traditionally approached by analyzing the network’s maximum confidence. In this paper we tackle closely related tasks: predicting the accuracy of a neural network across an entire dataset, rather than the accuracy over individual data points.

---

> > ### Author Response · Authors · 2024-11-20
> > **Response (2/2) to Reviewer DBUB**
> >
> > > *Have you tried comparing to using real images from the training dataset (for example, CIFAR10 images) as probes?*
> >
> > Generally, this setting does not allow using real training data as it provides extra supervision, though we agree it would still be interesting to test it. We provide an additional analysis where in-distribution data points (CIFAR10 images) are given as probes. Since our initial choice for synthetic data of INRs was already the exact in-distribution probes, their results remain unchanged for the INR tasks. However, as shown in the table below, in CNNs model zoos in-distribution probes work better than synthetic data, while ProbeGen outperforms both in all scenarios. We will add these results as part of our motivation section, as can be seen in our revised manuscript where changes are marked in yellow.
> >
> > | Method               | MNIST  | FMNIST | CIFAR10-GS | CIFAR10 Wild Park |
> > |----------------------|--------|--------|------------|--------------------|
> > | Vanilla Probing      | 0.955  | 0.808  | 0.936      | 0.889             |
> > | Synthetic Data       | 0.959  | 0.856  | 0.917      | 0.893             |
> > | In-Distribution Data |   0.959    |   0.856    | 0.942      | 0.930             |
> > | ProbeGen             | 0.984  | 0.877  | 0.957      | 0.932             |
> >
> >
> > ___
> >
> > > *[...] so it will be beneficial to provide results for more challenging setups with higher-dimensional input spaces (e.g., point clouds) or input spaces with more challenging structures (e.g., graphs).*
> >
> > While this is intriguing, all the standard weight-space benchmarks are for images and creating baselines in other modalities is not trivial. None-the-less we agree about the importance of such benchmarks and hope to explore this in future work.
> > ___
> >
> > > *While the authors analyze the computational cost, showing clear benefits compared to the GNN-based approach, it is not clear that the results will hold for larger models, as the approach requires forward and backward steps through the models.*
> >
> > As mentioned in our first rebuttal point, running graph based approaches on larger scale becomes impossible due to their quadratic neuron based scaling laws. While our approach requires multiple forward passes at test time (and backwards steps at train time), graph-based approaches scale much worse: They apply a deep network with several layers to each neuron and each weight of the model individually. This makes these approaches much less scalable.
> >
> > ___
> >
> > > *The Appendix is completely missing, along with additional important information regarding implementation details, experimental setup, HPs, etc.*
> >
> > We added an appendix including all the necessary implementation details and additional visualizations of the generated probes by different methods at the end of our revised manuscript.
> >
> > ___
> >
> > > *It is not clear if you also used Conv layers for the INR tasks. If so, why does this make sense?*
> >
> > We do not use a conv based generator in the INR experiments, instead we choose $2$ fully-connected layers as the generator. This works well in this case due to the simple structure of the data (Fig. 6). We added this to the implementation details in our revised manuscript.
> >
> > ___
> >
> > As far as we understand, the main issues the reviewer had with the paper is regarding the lack of larger-scaled experiments. We hope we were able to alleviate this concern with our added ResNets experiment. If the reviewer has additional questions or comments, we will be happy to address them during the discussion period. If we were able to address the concerns, we ask that the reviewer consider increasing their rating.

---

> > > ### Author Response · Authors · 2024-11-23
> > > **Follow-up**
> > >
> > > Thank you again for your time and effort in reviewing our work. We are following up to kindly ask if you had the chance to review our response submitted on November 20?
> > >
> > > If there are any remaining questions or concerns, we would be happy to further discuss them. If our responses have addressed your concerns, we would greatly appreciate your reconsideration of the score.
> > >
> > > Thanks,
> > >
> > > The Authors

---

> > > > ### Comment · Reviewer_DBUB · 2024-11-25
> > > > **Response to Authors**
> > > >
> > > > I want to thank the authors for their response and for providing additional results and comparisons. I have gone through the reviews and the authors' responses. Some of my concerns were addressed, and so I will update my score to 6. I believe the paper will be much stronger and more convincing if the authors will include experimental results with higher-dimensional input spaces (e.g., point clouds), and/or input spaces with more challenging structures (e.g., graphs), and/or results for additional data modalities. As stated by reviewer PoPY, there are existing benchmarks for that. I recommend that the authors include such experiments for the CR version.

---

> > > > > ### Author Response · Authors · 2024-11-26
> > > > >
> > > > > We thank the reviewer for the thoughtful feedback and recognition of our work. We agree that incorporating the proposed evaluation on point cloud trained networks, could further strengthen the applicability of ProbeGen. Therefore, as suggested by the reviewer and reviewer PoPY, we conducted an experiment on the Neural Field Arena [1], classifying INRs of point clouds from the ShapeNet dataset:
> > > > >
> > > > >
> > > > > | Method            |   Acc. |
> > > > > |----------------|-------|
> > > > > | Neural Graphs    |  0.903 |
> > > > > | DWS    |  0.911 |
> > > > > | Vanilla Probing (128)   |  0.913 |
> > > > > | ProbeGen (128)  |  0.930 |
> > > > >
> > > > >
> > > > > The results clearly demonstrate that ProbeGen outperforms all prior approaches, including Vanilla Probing. Moreover, we conducted an experiment where we test ProbeGen and Vanilla Probing with more probes. The table below shows that ProbeGen surpasses all baselines even with a highly restrictive budget of $32$. We highlight, that this $32$ ProbeGen version outperforms a $2048$ probe Vanilla Probing, showing ProbeGen's better scaling laws.
> > > > >
> > > > >
> > > > > | Probes                | Vanilla Probing     | ProbeGen          |
> > > > > |-----------------------|---------------------|-------------------|
> > > > > | 32                    |        0.879        |  0.928            |
> > > > > | 64                    | 0.903               |     0.927         |
> > > > > | 128                   | 0.913               |     0.930         |
> > > > > | 256                   | 0.918               |     0.932         |
> > > > > | 512                   | 0.921               |     0.932         |
> > > > > | 1024                  | 0.924               |      0.931        |
> > > > > | 2048                  | 0.925               |      0.931        |
> > > > >
> > > > >
> > > > > We hope these additional results address the reviewer’s concerns and further validate the contributions of our work.
> > > > >
> > > > > [1] Papa et al. How to Train Neural Field Representations: A Comprehensive Study and Benchmark. CVPR 2024

---

### Official Review · Reviewer_4rBX · 2024-10-31

**Soundness:** 3
**Presentation:** 4
**Contribution:** 2
**Rating:** 6
**Confidence:** 4

**Summary:**

This paper advocates for probing methods to solve tasks on the weight space of NNs. It first shows that naive probing performs as well as SoTA methods that operate directly on the weight space in discriminative tasks. Then, by analyzing the learned probs, the authors suggest an improved mechanism for learning the probs using deep linear networks which adds implicit regularization. The authors demonstrate the merits of their approach compared to baseline methods on two tasks, INR classification and image prediction errors. Several ablations are done to inspect different aspects of the method.

**Strengths:**

* The idea of using a deep linear network to obtain implicit regularization while guarding against potential overfitting is nice and original in that context.
* The proposed approach is extremely more efficient compared to methods that learn directly on the weights.
* The paper is written clearly and adequately. The authors empirically justify their design choices every step of the way and show other alternatives (such as learned vs unlearned probs, and the architecture decisions about the generator).
* The ablation study clears several ambiguities that arise while reading the paper, such as using non-linear generators and the importance of the inductive biases introduced into it.

**Weaknesses:**

* The authors discussed frankly about the main limitations of their approach, which I agree with. In particular, there are some tasks on which it is not immediately apparent how to use linear probing, such as generative tasks. It may be valuable if the authors could come up with possible remedies for that, but I understand if they don't since it is beyond the scope of this work and maybe non-trivial.

* Experiments:
  * Section 5.2 demonstrates the importance of deep linear classifiers compared to non-linear ones. The main issue, as I understand it, is overfitting. But, I wonder why didn't you try explicit regularization schemes to benefit from a more expressive network? I believe that an experiment of that flavor is missing.
  * Given the main advantage of this approach (i.e., agnostic to symmetries and fast) and given its limited applicability, I believe the submission can be made much stronger by evaluating ProbeGen on complex data, and on bigger networks. Both are limitations of current models that operate on the weights directly. For instance, INR classification or generalization error of images from ImageNet.
  * Related to my previous comment - I can see why on MNIST and even CIFAR small number of probs is sufficient to get good results. However, I speculate that it will not be the case for more challenging datasets (e.g., ImageNet or datasets of fine-grained classification such as CUB). Perhaps additional experiments can clarify that ambiguity.
  * Exact experimental details are missing to reproduce the results, e.g., what optimizers were used? Did you apply regularization? What is the architecture of the classifier? Also, it is not clear whether data augmentation was used for baseline methods (as it tends to improve the results).

* Some parts of the submission can be improved. Specifically,
  * There should be some background on INRs and the tasks involving them.
  * It is not clear how exactly Kendall’s $\tau$ was used to evaluate the generalization error.
  * It is not clear how exactly to run the code in the submission (there are no explanations about how to obtain the data, some files seem to be missing, etc.)

**Questions:**

.

---

> ### Author Response · Authors · 2024-11-20
> **Response (1/2) to Reviewer 4rBX**
>
> We thank the reviewer for their effort and for finding our technical deep linear generators “original in that context”. Below, we address the reviewer’s concerns in detail.
> ___
>
> > *Given the main advantage of this approach (i.e., agnostic to symmetries and fast) and given its limited applicability, I believe the submission can be made much stronger by evaluating ProbeGen on complex data, and on bigger networks....*
>
> We conducted a larger scale experiment on ResNet18. The largest Model Zoo originally evaluated in our paper, contains models with 17.5K parameters on average. **In contrast, our ResNet18 models have over 11.1M parameters, more than a 600x scale up.** We trained over $6,000$ ResNet18 models using different subsets of Tiny-ImageNet, where the weight learning task is to predict their test accuracy. Each subset is of $10$ classes, and there are $10$ different subsets in total.
>
> The results are presented in the table below. ProbeGen scaled to larger models, reaching a kendall's $\tau$ of $0.91$ compared to statNN and Vanilla Probing which achieve $0.86$ and $0.843$ respectively. Neural-Graphs were too computationally expensive to scale to this experiment. We will release this new Model Zoo to encourage further research in this area.
>
> | Method           | Kendall's $\tau$          |
> |------------------|---------------------------|
> | StatNN           |          0.860            |
> | Neural Graphs    |            -              |
> | Vanilla Probing  |          0.843            |
> | ProbeGen         |          **0.910**        |
>
> This experiment demonstrates that ProbeGen can effectively scale to models with millions of parameters.
>
> ___
>
> > *[...] But, I wonder why didn't you try explicit regularization schemes to benefit from a more expressive network?...*
>
> We include this experiment here, comparing deep linear generators to a range of L2 weight decay regularization strengths. We present the results in the table below. Our results show that while adding some regularization might be helpful, it is not as good as a deep linear generator. Moreover, explicit regularization requires aggressive hyper-parameter tuning of the regularization strength, including testing non-standard values (a weight decay of $10^{-7},10^{-8}$), while deep linear generators do not. Moreover, these regularizations are not consistent between experiments, and there is no single value that fits all experiments. We conclude that a deep linear generator, as presented in ProbeGen, provides a stable and subtle regularization that fits weight-space analysis.
>
>
> | Method                                      | MNIST  | FMNIST  | CIFAR10-GS | CIFAR10 Wild Park |
> |---------------------------------------------|--------|---------|------------|--------------------|
> | Non-Linear Generator + L2 Reg. ($10^{-2}$ WD) | 0.942  | 0.531   | 0.758      | 0.442             |
> | Non-Linear Generator + L2 Reg. ($10^{-3}$ WD) | 0.979  | 0.858   | 0.824      | 0.547             |
> | Non-Linear Generator + L2 Reg. ($10^{-4}$ WD) | 0.981  | 0.867   | 0.861      | 0.737             |
> | Non-Linear Generator + L2 Reg. ($10^{-5}$ WD) | 0.982  | 0.869   | 0.911      | 0.871             |
> | Non-Linear Generator + L2 Reg. ($10^{-6}$ WD) | 0.982  | 0.870   | 0.941      | 0.905             |
> | Non-Linear Generator + L2 Reg. ($10^{-7}$ WD) | 0.982  | 0.869   | 0.947      | 0.922             |
> | Non-Linear Generator + L2 Reg. ($10^{-8}$ WD) | 0.981  | 0.870   | 0.948      | 0.928             |
> | ProbeGen                                    | 0.984  | 0.877   | 0.957      | 0.932             |
>
> We included these results in our revised manuscript.

---

> > ### Author Response · Authors · 2024-11-20
> > **Response (2/2) to Reviewer 4rBX**
> >
> > > *[...] there are some tasks on which it is not immediately apparent how to use linear probing, such as generative tasks. It may be valuable if the authors could come up with possible remedies for that, but I understand if they don't since it is beyond the scope of this work and maybe non-trivial.*
> >
> > Indeed, our approach focuses on discriminative rather than generative weight-space tasks. However, we stress that discriminative tasks are important, have a very wide scope, and we barely scratched the surface. For instance, fields like membership inference attacks and model calibration aim to predict a network’s accuracy for specific samples, and are traditionally approached by analyzing the network’s maximum confidence. In this paper we tackle closely related tasks: predicting the accuracy of a neural network across an entire dataset, rather than the accuracy over individual data points.
> >
> > ___
> >
> > > *[...] I can see why on MNIST and even CIFAR small number of probs is sufficient to get good results. However, I speculate that it will not be the case for more challenging datasets*
> >
> > We agree with the reviewer that in more challenging datasets, with more classes, these scaling laws might differ in numbers, i.e., more probes will be required for achieving good performance. However, the main trend should remain the same: a sharp rise followed by diminishing gains.
> >
> > ___
> >
> > > *Exact experimental details are missing to reproduce the results...*
> >
> > We added an appendix to our revised manuscript with these implementation details. Specifically, about our baselines: we ran all baselines as intended by their authors, meaning augmentations were included when they were included by the authors code.
> >
> > ___
> >
> > > *There should be some background on INRs and the tasks involving them.*
> >
> > We added more background in our revised manuscript, within the related works. Changes are marked in yellow.
> >
> > ___
> >
> > > *It is not clear how exactly Kendall’s τ was used to evaluate the generalization error.*
> >
> > In our revised manuscript, we included a description of the kendall's $\tau$ at our motivation section as well as our experimental section. Changes are marked in yellow.
> >
> > ___
> >
> > > *It is not clear how exactly to run the code in the submission (there are no explanations about how to obtain the data, some files seem to be missing, etc.)*
> >
> > We will upload a revised version of the code, with detailed documentation and instructions on how to get the datasets.
> >
> > ___
> >
> > As far as we understand, the main issues the reviewer had with the paper is regarding the regularization claim, lack of larger-scaled experiments, and performance of in-distribution data probes. We hope we were able to alleviate all of these concerns with our ResNets experiment, regularization ablation and in-distribution ablation. If the reviewer has additional questions or comments, we will be happy to address them during the discussion period. If we were able to address the concerns, we ask that the reviewer consider increasing their rating.

---

> > > ### Author Response · Authors · 2024-11-23
> > > **Follow-up**
> > >
> > > Thank you again for your time and effort in reviewing our work. We are following up to kindly ask if you had the chance to review our response submitted on November 20?
> > >
> > > If there are any remaining questions or concerns, we would be happy to further discuss them. If our responses have addressed your concerns, we would greatly appreciate your reconsideration of the score.
> > >
> > > Thanks,
> > >
> > > The Authors

---

> > > > ### Comment · Reviewer_4rBX · 2024-11-24
> > > >
> > > > I thank the authors for the answers to my comments. After reading all the reviews and the authors' responses I decided to raise my score to 6, mainly because of the new benchmark on large models which I believe strengthens the submission.

---

> > > > > ### Author Response · Authors · 2024-11-25
> > > > >
> > > > > Thank you for considering our rebuttal and increasing your score. We truly appreciate your thoughtful feedback and recognition of our work.

---

### Official Review · Reviewer_PoPY · 2024-11-01

**Soundness:** 3
**Presentation:** 3
**Contribution:** 3
**Rating:** 6
**Confidence:** 4

**Summary:**

This work addresses the task of learning in weight spaces, and introduces a probing-based method that passes a set of learnable inputs to a model and trains a simple network on top of the concatenated probe outputs. The authors first demonstrate that standalone vanilla probing performs very well, almost on par to state-of-the-art methods. They then extend the method by introducing a shared deep linear network module, and incorporate inductive biases towards structured probes. This module further increases performance by reducing overfitting, while being more efficient than other approaches.

**Strengths:**

The paper is well written and is quite easy to follow. The connections with binary code analysis and the corresponding insights are very useful towards more scalable weight-space methods.

The proposed method reaches state-of-the-art performance with high computational efficiency. The authors provide a comprehensive set of ablation studies that offers significant insights on probing-based methods for weight space learning.

**Weaknesses:**

The proposed method cannot be used in weight-level tasks, such as editing neural network weights, which represents a large class of tasks in learning in weight spaces. The impact of this work would greatly increase if the authors include a more detailed discussion on why that is the case, and potential directions to alleviate this limitation.

It is unclear if the proposed method can be used to complement existing mechanistic approaches and result in a performance greater than either of the two components. See question 1.

**Questions:**

[1] It seems intuitive that probing approaches are complementary to mechanistic approaches. Can the proposed method be combined with existing mechanistic approaches (e.g. Neural Graphs)? Does ProbeGen increase the performance of these methods? An ablation study on whether deep linear generators with structured input probes boost mechanistic approaches would be very useful, and would further increase the impact of the proposed method.

[2] The experiments on learned vs unlearned probes are very insightful. It is unclear, though, why the authors did not explore actual unlearned images as probes, potentially in-distribution. For example, a small subset of the training (or validation) set used only for probing. What is the impact of using real images for probes?

[3] Deep linear networks are introduced as a means to introduce regularization and reduce overfitting. Have the authors explored other common regularization methods, e.g. adding weight decay, or dropout? Is the reduced expressivity of the network an advantage here? If so, it is not very clear why that is the case.

[4] The authors often mention the terms "training dataset classification", "extract information about its training dataset", or variants. These terms seem quite confusing. Do they refer to INR classification tasks? The manuscript could benefit from some rephrasing in these instances.

[5] As the number of probes increases, it approaches the size of the image for INRs, or the test set size for CNN generalization.
In that case, probing in ProbeGen becomes almost trivial, since it is nearly identical to feeding the whole image to an MLP, or to running the CNN over the whole test set. In both cases, we can expect the performance monotonically increasing as a function of the number of probes. Is that assumption correct? Is there a sweet spot between efficiency and performance? An ablation study on MNIST/Fashion MNIST with increasing numbers of probes (up to the image size) that shows performance and FLOPS would be very useful here.

---

> ### Author Response · Authors · 2024-11-20
> **Response (1/2) to Reviewer PoPY**
>
> We thank the reviewer for their effort and for finding our “connections with binary code analysis and the corresponding insights” to be “very useful towards more scalable weight-space methods”. Below, we address the reviewer’s concerns in detail.
> ___
>
> > *It is unclear if the proposed method can be used to complement existing mechanistic approaches....* + *[...] An ablation study on whether deep linear generators with structured input probes boost mechanistic approaches would be very useful, and would further increase the impact of the proposed method.*
>
> We tested this by adding a deep linear generator module to the Neural-Graphs baseline. The results, presented in the table below, show that our generator module enhances the performance of Neural-Graphs significantly. We did not find that using Graphs was better than our standard ProbeGen approach.
>
> | Method                     | CIFAR10-GS | CIFAR10 Wild Park |
> |----------------------------|------------|--------------------|
> | Neural Graphs              | 0.938      | 0.885             |
> | Neural Graphs + ProbeGen   | 0.946      | 0.913             |
> | ProbeGen                   | 0.957      | 0.932             |
>
>
> ___
>
> > *[...] It is unclear, though, why the authors did not explore actual unlearned images as probes, potentially in-distribution…*
>
> Generally, this setting does not allow using real training data as it provides extra supervision, though we agree it would still be interesting to test it. We provide an additional analysis where in-distribution data points (CIFAR10 images) are given as probes. Since our initial choice for synthetic data of INRs was already the exact in-distribution probes, their results remain unchanged for the INR tasks. However, as shown in the table below, in CNNs model zoos in-distribution probes work better than synthetic data, while ProbeGen outperforms both in all scenarios. We will add these results as part of our motivation section, as can be seen in our revised manuscript where changes are marked in yellow.
>
> | Method               | MNIST  | FMNIST | CIFAR10-GS | CIFAR10 Wild Park |
> |----------------------|--------|--------|------------|--------------------|
> | Vanilla Probing      | 0.955  | 0.808  | 0.936      | 0.889             |
> | Synthetic Data       | 0.959  | 0.856  | 0.917      | 0.893             |
> | In-Distribution Data |   0.959    |   0.856    | 0.942      | 0.930             |
> | ProbeGen             | 0.984  | 0.877  | 0.957      | 0.932             |
>
> ___
>
> > *Deep linear networks are introduced as a means to introduce regularization and reduce overfitting. Have the authors explored other common regularization methods, e.g. adding weight decay…*
>
> We include this experiment here, comparing deep linear generators to a range of L2 weight decay regularization strengths. We present the results in the table below. Our results show that while adding some regularization might be helpful, it is not as good as a deep linear generator. Moreover, explicit regularization requires aggressive hyper-parameter tuning of the regularization strength, including testing non-standard values (a weight decay of $10^{-7},10^{-8}$), while deep linear generators do not. Moreover, these regularizations are not consistent between experiments, and there is no single value that fits all experiments. We conclude that a deep linear generator, as presented in ProbeGen, provides a stable and subtle regularization that fits weight-space analysis.
>
> | Method                                      | MNIST  | FMNIST  | CIFAR10-GS | CIFAR10 Wild Park |
> |---------------------------------------------|--------|---------|------------|--------------------|
> | Non-Linear Generator + L2 Reg. ($10^{-2}$ WD) | 0.942  | 0.531   | 0.758      | 0.442             |
> | Non-Linear Generator + L2 Reg. ($10^{-3}$ WD) | 0.979  | 0.858   | 0.824      | 0.547             |
> | Non-Linear Generator + L2 Reg. ($10^{-4}$ WD) | 0.981  | 0.867   | 0.861      | 0.737             |
> | Non-Linear Generator + L2 Reg. ($10^{-5}$ WD) | 0.982  | 0.869   | 0.911      | 0.871             |
> | Non-Linear Generator + L2 Reg. ($10^{-6}$ WD) | 0.982  | 0.870   | 0.941      | 0.905             |
> | Non-Linear Generator + L2 Reg. ($10^{-7}$ WD) | 0.982  | 0.869   | 0.947      | 0.922             |
> | Non-Linear Generator + L2 Reg. ($10^{-8}$ WD) | 0.981  | 0.870   | 0.948      | 0.928             |
> | ProbeGen                                    | 0.984  | 0.877   | 0.957      | 0.932             |
>
>
> We included these results in our revised manuscript.

---

> > ### Author Response · Authors · 2024-11-20
> > **Response (2/2) to Reviewer PoPY**
> >
> > > *The proposed method cannot be used in weight-level tasks, such as editing neural network weights, which represents a large class of tasks in learning in weight spaces...*
> >
> > Indeed, our approach focuses on discriminative rather than generative weight-space tasks. However, we stress that discriminative tasks are important, have a very wide scope, and we barely scratched the surface. For instance, fields like membership inference attacks and model calibration aim to predict a network’s accuracy for specific samples, and are traditionally approached by analyzing the network’s maximum confidence. In this paper we tackle closely related tasks: predicting the accuracy of a neural network across an entire dataset, rather than the accuracy over individual data points.
> >
> > ___
> >
> > > *[...] An ablation study on MNIST/Fashion MNIST with increasing numbers of probes (up to the image size) that shows performance and FLOPS would be very useful here.*
> >
> > We agree with the reviewer and provide the experiment here, gradually increasing the number of optimized probes up to $1024$ probes in the FMNIST dataset. While the results show improved performance with additional probes, the gains diminish as the number of probes increases. Based on these findings, we believe there is a sweet-spot where performance is near optimal, yet requires considerably less computational efforts.
> >
> > | Method          | FMNIST  | FMNIST-FLOPs |
> > |------------------------|---------|--------------|
> > | ProbeGen (16 Probes)    | 0.805   | 8.619M       |
> > | ProbeGen (32 Probes)    | 0.844   | 8.767M       |
> > | ProbeGen (64 Probes)    | 0.861   | 9.064M       |
> > | ProbeGen (128 Probes)   | 0.877   | 9.658M       |
> > | ProbeGen (256 Probes)   | 0.888   | 10.846M      |
> > | ProbeGen (512 Probes)   | 0.888   | 13.222M      |
> > | ProbeGen (1024 Probes)  | 0.888   | 17.973M      |
> >
> > ___
> >
> > > *The authors often mention the terms "training dataset classification", "extract information about its training dataset", or variants. These terms seem quite confusing. Do they refer to INR classification tasks?*
> >
> > Yes, we clarified these in our revised manuscript.

---

> > > ### Author Response · Authors · 2024-11-23
> > > **Follow-up**
> > >
> > > Thank you again for your time and effort in reviewing our work. We are following up to kindly ask if you had the chance to review our response submitted on November 20?
> > >
> > > If there are any remaining questions or concerns, we would be happy to further discuss them. If our responses have addressed your concerns, we would greatly appreciate your reconsideration of the score.
> > >
> > > Thanks,
> > >
> > > The Authors

---

> > > > ### Comment · Reviewer_PoPY · 2024-11-24
> > > > **Comment by Reviewer PoPY**
> > > >
> > > > I would like to thank the authors for a detailed rebuttal that addressed most of my concerns. I think this is a good paper and I maintain my score. As a final note, following up on Reviewer's DBUB question on "higher-dimensional input spaces (e.g., point clouds)", I would recommend that the authors include such experiments for the camera-ready version of the paper, noting that there are existing benchmarks for shapes (ShapeNet10) from NFN and Fit-a-Nef [1].
> > > >
> > > > [1] Papa et al. How to Train Neural Field Representations: A Comprehensive Study and Benchmark. CVPR 2024

---

> > > > > ### Author Response · Authors · 2024-11-26
> > > > >
> > > > > We thank the reviewer for the thoughtful feedback and recognition of our work. We agree that incorporating the proposed evaluation on point cloud trained networks, could further strengthen the applicability of ProbeGen. Therefore, as suggested by the reviewer and reviewer DBUB, we conducted an experiment on the Neural Field Arena [1], classifying INRs of point clouds from the ShapeNet dataset:
> > > > >
> > > > >
> > > > > | Method            |   Acc. |
> > > > > |----------------|-------|
> > > > > | Neural Graphs    |  0.903 |
> > > > > | DWS    |  0.911 |
> > > > > | Vanilla Probing (128)   |  0.913 |
> > > > > | ProbeGen (128)  |  0.930 |
> > > > >
> > > > >
> > > > > The results clearly demonstrate that ProbeGen outperforms all prior approaches, including Vanilla Probing. Moreover, we conducted an experiment where we test ProbeGen and Vanilla Probing with more probes. The table below shows that ProbeGen surpasses all baselines even with a highly restrictive budget of $32$. We highlight, that this $32$ ProbeGen version outperforms a $2048$ probe Vanilla Probing, showing ProbeGen's better scaling laws.
> > > > >
> > > > >
> > > > > | Probes                | Vanilla Probing     | ProbeGen          |
> > > > > |-----------------------|---------------------|-------------------|
> > > > > | 32                    |        0.879        |  0.928            |
> > > > > | 64                    | 0.903               |     0.927         |
> > > > > | 128                   | 0.913               |     0.930         |
> > > > > | 256                   | 0.918               |     0.932         |
> > > > > | 512                   | 0.921               |     0.932         |
> > > > > | 1024                  | 0.924               |      0.931        |
> > > > > | 2048                  | 0.925               |      0.931        |
> > > > >
> > > > >
> > > > > We hope these additional results address the reviewer’s concerns and further validate the contributions of our work.
> > > > >
> > > > > [1] Papa et al. How to Train Neural Field Representations: A Comprehensive Study and Benchmark. CVPR 2024

---

> > > > > > ### Comment · Reviewer_PoPY · 2024-11-26
> > > > > > **Comment by Reviewer PoPY**
> > > > > >
> > > > > > I would like to thank the authors for their response, and I am happy to see that all other reviewers have raised their score. This experiment, in my opinion, solidifies the contributions of this work further. I encourage the authors to include it in the camera-ready version of the paper.
> > > > > >
> > > > > > One final note (not necessarily expecting a response), I find it very surprising that the performance of both Vanilla Probing and ProbeGen is this high even with 32 or 64 probes, and that ProbeGen only becomes marginally better with the addition of more probes.

---

### Official Review · Reviewer_wSNk · 2024-11-02

**Soundness:** 2
**Presentation:** 3
**Contribution:** 2
**Rating:** 6
**Confidence:** 5

**Summary:**

In this paper, the authors introduce ProbeGen, a deep linear method designed for probing model data in weight space learning. They begin by demonstrating empirically that probing is an effective approach for weight space learning, although it has limitations and room for improvement. Building on these findings, they propose ProbeGen as an enhancement and subsequently validate its effectiveness through a series of experiments.

**Strengths:**

- This work addresses the novel and intriguing field of weight space learning, aiming to tackle the significant challenge of simplifying optimization within this domain.
- The proposed method ProbeGen is simple, easy to follow, and yet effective.
- The authors empirically demonstrate the motivation behind ProbeGen, providing interesting insights.
- The paper is well-written and easy to follow.

**Weaknesses:**

- The experimental part is limited. From an empirical-oriented paper, I expect the experimental section to be more comprehensive. For instance, the effectiveness of ProbeGen is shown only on INRs overlooking equivariant tasks in weight space learning like domain adaptation [1], alignment [2], editing [3], and more.
- The authors mentioned that ProbeGen could be used for processing high-dimensional inputs, however, the experimental section focuses on small-scale INRs instead of modern architectures.
- It would be interesting to see what is the upper bound of ProbeGen w.r.t number of probes. The authors used a max number of 128 probes what will be the performance if we will use 1024 probes for example? Can we match the image space performance?
- The authors failed to cite relevant works in the field of weight space learning [2,4,5,6].
- Not sure if it is intended but the supplementary material is missing.

------------

[1] Equivariant Architectures for Learning in Deep Weight Spaces, Navon et al.

[2] Equivariant Deep Weight Space Alignment, Navon et al.

[3] Permutation equivariant neural functionals, Zhou et al.

[4] Improved Generalization of Weight Space Networks via Augmentations, Shamsian et al.

[5] Equivariant Neural Functional Networks for Transformers, Tran et al.

[6] Learning on LoRAs: GL-Equivariant Processing of Low-Rank Weight Spaces for Large Finetuned Models, Putterman et al.

**Questions:**

- Do the authors have a hypothesis as to why non-linear activation functions lead to poorer probe representations? Have the authors experimented with other non-linear activations besides ReLU activations discussed in lines 255-261?

- It would be interesting to try and enrich the probes by applying augmentations to the input like presented in [1,4].

---

> ### Author Response · Authors · 2024-11-20
> **Response (1/2) to Reviewer wSNk**
>
> We thank the reviewer for recognizing that our method is “easy to follow, and yet effective” and that the motivation behind it provides “interesting insights”. Below, we address the reviewer’s concerns in detail.
>
> ___
>
> > *[...] the experimental section focuses on small-scale INRs instead of modern architectures.*
>
> We conducted a larger scale experiment on ResNet18. The largest Model Zoo originally evaluated in our paper, contains models with 17.5K parameters on average. **In contrast, our ResNet18 models have over 11.1M parameters, more than a 600x scale up.** We trained over $6,000$ ResNet18 models using different subsets of Tiny-ImageNet, where the weight learning task is to predict their test accuracy. Each subset is of $10$ classes, and there are $10$ different subsets in total.
>
> The results are presented in the table below. ProbeGen scaled to larger models, reaching a kendall's $\tau$ of $0.91$ compared to statNN and Vanilla Probing which achieve $0.86$ and $0.843$ respectively. Neural-Graphs were too computationally expensive to scale to this experiment. We will release this new Model Zoo to encourage further research in this area.
>
> | Method           | Kendall's $\tau$    |
> |------------------|---------------------------|
> | StatNN           |          0.860            |
> | Neural Graphs    |            -              |
> | Vanilla Probing  |          0.843         |
> | ProbeGen         |          **0.910**     |
>
> This experiment demonstrates that ProbeGen can effectively scale to models with millions of parameters.
>
> ___
>
> > *[...] It would be interesting to see what is the upper bound of ProbeGen w.r.t number of probes [...] Can we match the image space performance?*
>
> We ablate the upper bound w.r.t the number of probes using the FMNIST dataset. While the results show improved performance with additional probes, the gains diminish as the number of probes increases. Based on these findings, we believe there is a sweet-spot for choosing the number of probes, where performance is near optimal yet it requires considerably less computational efforts.
>
> | Method          | FMNIST  | FMNIST-FLOPs |
> |------------------------|---------|--------------|
> | ProbeGen (16 Probes)    | 0.805   | 8.619M       |
> | ProbeGen (32 Probes)    | 0.844   | 8.767M       |
> | ProbeGen (64 Probes)    | 0.861   | 9.064M       |
> | ProbeGen (128 Probes)   | 0.877   | 9.658M       |
> | ProbeGen (256 Probes)   | 0.888   | 10.846M      |
> | ProbeGen (512 Probes)   | 0.888   | 13.222M      |
> | ProbeGen (1024 Probes)  | 0.888   | 17.973M      |
>
>
> In terms of image-level performance, we train a simple convolutional image-level classifier, and find it to perform better ($91.6\%$ accuracy) as it includes priors specific to images. However, when using a similar classifier to ProbeGen with Fully-Connected layers instead, image-level performance is $88.8\%$ similar to ProbeGen with $256$ probes. This means our generator finds good probes for this task, being able to match image-level performance, with a third the number of pixels available.
>
> ___
>
> > *[...] the effectiveness of ProbeGen is shown only on INRs overlooking equivariant tasks in weight space learning like domain adaptation [1], alignment [2], editing [3], and more*
>
> Indeed, our approach focuses on discriminative rather than generative weight-space tasks. However, we stress that discriminative tasks are important, have a very wide scope, and we barely scratched the surface. For instance, fields like membership inference attacks and model calibration aim to predict a network’s accuracy for specific samples, and are traditionally approached by analyzing the network’s maximum confidence. In this paper we tackle closely related tasks: predicting the accuracy of a neural network across an entire dataset, rather than the accuracy over individual data points.

---

> > ### Author Response · Authors · 2024-11-20
> > **Response (2/2) to Reviewer wSNk**
> >
> > > *[...] Have the authors experimented with other non-linear activations besides ReLU activations discussed in lines 255-261?*
> >
> > We ablate the use of the non-linear generator with different activations. The results are presented in the table below. We can see that while some activations are indeed better than ReLU, none of them is better than ProbeGen. Moreover, it is non-trivial which activation should be chosen for each task, while ProbeGen presents a single simple alternative. This indicates that our linear generator indeed serves as a helpful regularization.
> >
> > | Experiment 6              | MNIST  | FMNIST  | CIFAR10-GS | CIFAR10 Wild Park |
> > |---------------------------|--------|---------|------------|--------------------|
> > | Generator w. ReLU         | 0.982  | 0.871   | 0.949      | 0.928             |
> > | Generator w. Leaky-ReLU   | 0.982  | 0.872   | 0.950      | 0.934             |
> > | Generator w. GeLU         | 0.983  | 0.868   | 0.954      | 0.929             |
> > | Generator w. Sigmoid      | 0.978  | 0.851   | 0.904      | 0.867             |
> > | ProbeGen                  | 0.984  | 0.877   | 0.957      | 0.932             |
> >
> >
> > ___
> >
> > > *The authors failed to cite relevant works in the field of weight space learning [2,4,5,6].*
> >
> > We added all of these to our related work section, and will include some of them as baselines to our approach. The modified text and table is in our revised manuscript, where changes are marked in yellow.
> >
> > ___
> >
> > > *Not sure if it is intended but the supplementary material is missing.*
> >
> > We added an appendix including the necessary implementation details, as well as additional ablations and visualizations of generated probes by different methods. This can be seen at the end of our revised manuscript.  Please let us know if you find anything is missing from the appendix.
> >
> > ___
> >
> > > *It would be interesting to try and enrich the probes by applying augmentations to the input like presented in [1,4].*
> >
> > We tested adding the neural augmentations from [7], the results are below. We find it to hurt the performance of ProbeGen. We hypothesize that mechanistic approaches benefit from subtle augmentations to the weights, as they directly observe the weights. However, this type of augmentation is unfit for probing based methods as probing observes the functionality of a model, which could change unexpectedly when using this type of augmentations.
> >
> > | Method                   | FMNIST  |
> > |--------------------------|---------|
> > | ProbeGen                 | 0.877   |
> > | ProbeGen + Augmentations | 0.764   |
> >
> > [7] Kofinas, Miltiadis, et al. "Graph neural networks for learning equivariant representations of neural networks." arXiv preprint arXiv:2403.12143 (2024).
> > ___
> >
> > As far as we understand, the main issues the reviewer had with the paper is regarding the limited experimental section and lack of larger-scaled experiments. We hope we were able to clarify our focus on discriminative tasks, and alleviate the concern on larger-scales with our ResNets experiment. If the reviewer has additional questions or comments, we will be happy to address them during the discussion period. If we were able to address the concerns, we ask that the reviewer consider increasing their rating.

---

> > > ### Author Response · Authors · 2024-11-23
> > > **Follow-up**
> > >
> > > Thank you again for your time and effort in reviewing our work. We are following up to kindly ask if you had the chance to review our response submitted on November 20?
> > >
> > > If there are any remaining questions or concerns, we would be happy to further discuss them. If our responses have addressed your concerns, we would greatly appreciate your reconsideration of the score.
> > >
> > > Thanks,
> > >
> > > The Authors

---

> > > > ### Comment · Reviewer_wSNk · 2024-11-25
> > > > **Reviewer Response**
> > > >
> > > > Thanks to the authors for putting so much time and energy into this rebuttal. I believe that this paper is much stronger and comprehensive after the discussion period. After reading the other reviews and the authors responses I'm happy to align my score and raise it to 6.

---

> > > > > ### Author Response · Authors · 2024-11-26
> > > > >
> > > > > Thank you for considering our rebuttal and increasing your score. We truly appreciate your thoughtful feedback and recognition of our work.

---

### Meta-Review · Area_Chair_UH7H · 2024-12-22

**Metareview:**

The paper proposes deep linear probe generators, so that to use probing as an alternative and perhaps more intuitive and better scaleable way of modelling the weight space of neural networks (with applications to predicting generalization error, comparing architectures etc). All reviewers are positive about the value of the paper, both in terms of novelty and comparisons. There are some concerns regarding whether the comparisons are made on sufficiently big datasets, and generally if the method can generalize to more complex datasets and tasks, but they all agree on the value.

**Additional Comments On Reviewer Discussion:**

Reviewers in the post-rebuttal phase expressed general agreement on the value of the paper, with a consensus leaning towards positive but tempered enthusiasm. On one hand, it was acknowledged that the contributions are significant for an ICLR publication, with extensive ablations validating the proposed method. Preliminary experiments were seen as promising, particularly in demonstrating scalability to higher dimensions and more complex data. On the other hand, concerns were raised about the method’s potential to generalize effectively to more complex datasets and tasks, reflecting cautious optimism about its broader applicability.

---

### Decision · Program_Chairs · 2025-01-22

Accept (Poster)